# The impact of genetically controlled splicing on exon inclusion and protein structure

**Jonah Einson**[1,2]*, **Mariia Minaeva**[3], **Faiza Rafi**[2,4], **Tuuli Lappalainen**[2,3,5]

**1** Department of Biomedical Informatics, Columbia University Irving Medical Center, New York, NY, United States of America, **2** New York Genome Center, New York, NY, United States of America, **3** Science for Life Laboratory, Department of Gene Technology, KTH Royal Institute of Technology, Stockholm, Sweden, **4** Department of Biotechnology, The City College of New York, New York, NY, United States of America, **5** Department of Systems Biology, Columbia University Irving Medical Center, New York, NY, United States of America

* jee2142@cumc.columbia.edu

## Abstract

Common variants affecting mRNA splicing are typically identified though splicing quantitative trait locus (sQTL) mapping and have been shown to be enriched for GWAS signals by a similar degree to eQTLs. However, the specific splicing changes induced by these variants have been difficult to characterize, making it more complicated to analyze the effect size and direction of sQTLs, and to determine downstream splicing effects on protein structure. In this study, we catalogue sQTLs using exon percent spliced in (PSI) scores as a quantitative phenotype. PSI is an interpretable metric for identifying exon skipping events and has some advantages over other methods for quantifying splicing from short read RNA sequencing. In our set of sQTL variants, we find evidence of selective effects based on splicing effect size and effect direction, as well as exon symmetry. Additionally, we utilize AlphaFold2 to predict changes in protein structure associated with sQTLs overlapping GWAS traits, highlighting a potential new use-case for this technology for interpreting genetic effects on traits and disorders.

## Introduction

Alternative splicing is a fundamental cellular process which greatly increases the diversity of transcript isoforms across tissues and cell types in eukaryotes. It is estimated that the human transcriptome has a 10 fold increase in the number of alternatively spliced transcripts, compared to approximately 20,000 protein-coding genes [1], of which almost all undergo alternative splicing [2, 3]. From an evolutionary perspective, splicing changes have driven phenotypic differences between closely related vertebrates in a relatively short amount of time, highlighting its importance in gene function [4]. Furthermore, mRNA splicing patterns can be influenced by genetic variation across individuals and populations, as repeatedly demonstrated by studies that link common variants to splicing changes through the mapping of splicing quantitative trait loci (sQTLs) [5–9].

**Data Availability Statement:** Raw and processed data files are available to download at https://zenodo.org/record/8137337 and at gtexportal.org. Individual level genetic data is necessary for QTL mapping, but cannot be shared publicly because of

NIH policy, since it contains potentially sensitive information that could be used to identify anonymous GTEx donors. You must request access via dbGAP if you require these data (https://www.ncbi.nlm.nih.gov/projects/gap/cgi-bin/study.cgi?study_id=phs000424.v8.p2). Protected data are available in the AnVIL repository. Accompanying code is available to review and download at https://github.com/jeinson/sqtl_manuscript

**Funding:** This work was supported by the National Institutes of Health grants R01GM122924, R01MH106842 (J.E. & T.L.), the Data Driven Life Science program by the Knut and Alice Wallenberg Foundation grant WASPDDLS21:080 (M.M. & T.L.) and the S. Jay Levy Foundation (https://www.ccny.cuny.edu/sjaylevy) (F.R.). Part of the computations were enabled by resources provided by the Swedish National Infrastructure for Computing (SNIC) at UPPMAX partially funded by the Swedish Research Council through grant agreement no. 2018-05973. (M.M. & T.L.) The funders had no role in study design, data collection and analysis, decision to publish, or preparation of the manuscript.

**Competing interests:** I have read the journal's policy and the authors of this manuscript have the following competing interests: T.L. is a paid advisor to GSK, Pfizer, Goldfinch Bio and Variant Bio, and has equity in Variant Bio. This does not alter our adherence to PLOS ONE policies on sharing data and materials.

While most studies use RNA sequencing data to capture splicing events, they critically differ in the computational methods used to quantify splicing. Measuring alternative splicing through short read RNA-seq data is non-trivial, and always requires some level of compromise depending on the goals of the study. Oftentimes, a study aims to catalog as many splicing events as possible to increase power to detect splicing QTLs and characterize the types of genetic variants that affect splicing. These methods often consider different types of events, such as exon skipping and 3'/5' end usage simultaneously, which reduces the overall interpretability of the splicing signal but may provide insights into mechanisms of individual splicing events [7, 8, 10]. In other studies, splicing is quantified by inferring levels of full transcripts [11–13]. While this approach produces a biologically relevant splicing readout regarding downstream transcriptome effects, it is limited by isoform annotation and quantification, which is challenging from short-read RNA-seq data.

Splicing QTLs are known to colocalize with GWAS signals and potentially explain a considerable proportion of heritability of complex diseases [14–16]. Changes in splicing that associate with traits are likely mostly driven by differences in amino acid sequences that affect the function of downstream protein products [17–20]. These changes can be systematically mapped to functional domains by utilizing large databases of resolved protein structures like UniProt [21], where multiple isoforms splicing isoforms are curated for about 5,000 genes. This resource can help reveal the types of splicing events that may be most relevant for trait colocalization.

Most recently, through the development of AlphaFold2 [22], estimating the protein structure of splicing isoforms where an experimentally resolved structure is unavailable has become substantially easier and more reliable. Now, one can simply provide an amino acid sequence from two splice-isoforms, and interpret what parts of the protein are affected and to what degree [23–25]. This is especially relevant where alternate usage of rare isoforms may play a role in trait or disease risk. To date, no study has deeply probed how changes in splicing driven by genetic variation impact the function of proteins, which could reveal the causal mechanism underlying trait associations.

In this project, we map splicing QTLs in the GTEx resource [26] using an interpretable splicing phenotype that measures exon skipping events from RNA-seq split read counts. While we detect fewer sQTLs than some of the alternative approaches [26], our sQTLs are more optimized for downstream interpretation of splicing effects and for analyzing properties of genetically controlled exons. Additionally, by mapping changes in exon inclusion, we can more easily probe how protein structure is affected by these alterations, both by interpreting resolved protein structures and by predicting new structures with and without an alternatively spliced exon. Throughout our study, we demonstrate how this approach can reveal relevant biology, and how contemporary protein structure prediction further contextualizes the importance of genetically regulated splicing.

## Results

### A simple splicing phenotype improves interpretability of splicing QTLs

To begin, we cataloged splicing QTLs in protein coding genes using splicing quantified with the Percent Spliced In (PSI or $\psi$) metric on an exon-by-exon basis as a molecular phenotype. PSI directly captures exon skipping events, but makes no inference about whole isoform usage or complex splicing, which is advantageous for our downstream application. We used bulk RNA-sequencing data across 18 tissues and whole genome sequencing was from the Genotype-Tissue Expression Project Version 8 (GTEx v8), and applied the methods for sQTL mapping from GTEx [26], but with splicing quantified with the PSI phenotype (See methods for

details). This set of variant-exon pairs are hence referred to as ψQTLs, with variants and target exons referred to as sVariants and sExons respectively. Across tissues, limiting to one sExon per gene, we identified between 698 and 2,021 genes with a significant ψQTL (Fig 1B), with the number of significant genes correlating with the number of samples available per tissue, as is typical in QTL studies [9, 26–28] ($p = 0.0177$, Adjusted $R^2 = 0.2604$, S1A Fig). Significant variant-gene pairs replicated well, with a median $\pi_1$ score [29] between discovery and replication tissues of $0.8134 \pm 0.103$ IQR (S1F Fig). In total, we cataloged fewer ψQTL per tissue than in the GTEx [26] main analysis, which uses the Leafcutter cluster phenotype to quantify splicing and map QTLs in the same dataset (S1H Fig). While Leafcutter [8] identifies more splicing events and finds more sQTLs, it presents an interpretability challenge. It is often difficult to identify which exon a Leafcutter cluster corresponds to, and effect directions are sometimes unclear. While ψQTLs are less powerful in a statistical sense, the method clearly links splicing events to exons, genes, and effect directions, which was advantageous to the purpose of this study.

As an additional follow up, we performed ψQTL mapping in the Geuvadis [30]dataset, and checked for concordance with GTEx EBV-transformed lymphocytes. (See methods for details) Of the 1,119 sGenes with coverage in Geuvadis and GTEx, 423 had a significant sExon in both. The coverage in Geuvadis was smaller overall, only capturing splicing for 853 out of 1431 exons with a significant ψQTL in GTEx lymphocytes. Among this set, however, the concordance was reasonably high, with a $\pi_1$ score of 0.63. This replication strengthens evidence that ψQTLs are robust in many contexts. In a similar follow up analysis, we checked for concordance between our ψQTLs and sQTLs reported in Garrido-Martín et al. [18], where splicing also detected in GTEx V8, but is defined as a ratio of transcript isoforms. Across 18 tissues, we were able to capture a median of 59.39% of significant sQTL variants in our dataset, and found that they replicated with a median $\pi_1$ score of $0.67 \pm 0.03$ IQR (S1G Fig). The unmapped variants had a MAF $< .05$, so were excluded from our analyses prior to this comparison. (See Methods for details).

Moving forward, we obtained a final set of GTEx ψQTLs for downstream analyses by collapsing ψQTLs across tissues, considering the tissue where the ψQTL had the highest effect size (ΔPSI) when it appeared in multiple tissues, and removing genes where the 3' or 5' terminal exon was the most significant exon. This filter focused our analyses on exon skipping events. In total, we obtained a set of 4,835 genes with a significant ψQTL. In comparison to other variably spliced exons from genes that lacked a ψQTL, sExons were slightly shorter in bp (Mean bp: 141 and 137 respectively, Mann-Whitney $U$-test p = 0.022, S1B Fig). Additionally, among ψQTLs, sExons were more likely to fall in the later part of the transcript ($\chi^2$ Uniformity test $p = 5.28 \ 10^{-173}$, S1C Fig), also when compared to the same set of variably spliced exons in genes with no ψQTL (Mann-Whitney $U$-test $p = 0.00115$, S2 Fig). This is consistent with the observation that splicing QTLs tend to be more active post-transcriptionally [18].

One advantage of our approach is that ψQTL analysis allows for direct evaluation of exon symmetry. Symmetry refers to whether an exon has a length in base pairs that is divisible by 3, and therefore encodes a complete reading frame. We hypothesize that ψQTLs induce changes in exon inclusion that have a relatively low impact on fitness, since sVariants by definition are common in the population. Non-symmetric exons almost always induce a frameshift when they are alternatively spliced [31, 32], so we predict that ψQTLs will be enriched for symmetric exons. We found that indeed, among sExons, 41.20% were symmetric (58.6% non-symmetric) compared to 38.77% of all non-terminal exons that were not sExons (61.23% non-symmetric) annotated in gencode v26 (Fig 1C, Fisher's Exact Test $p = 6.64 \times 10^{-4}$), providing evidence that common splice-regulatory variants are less likely to severely impact gene function.

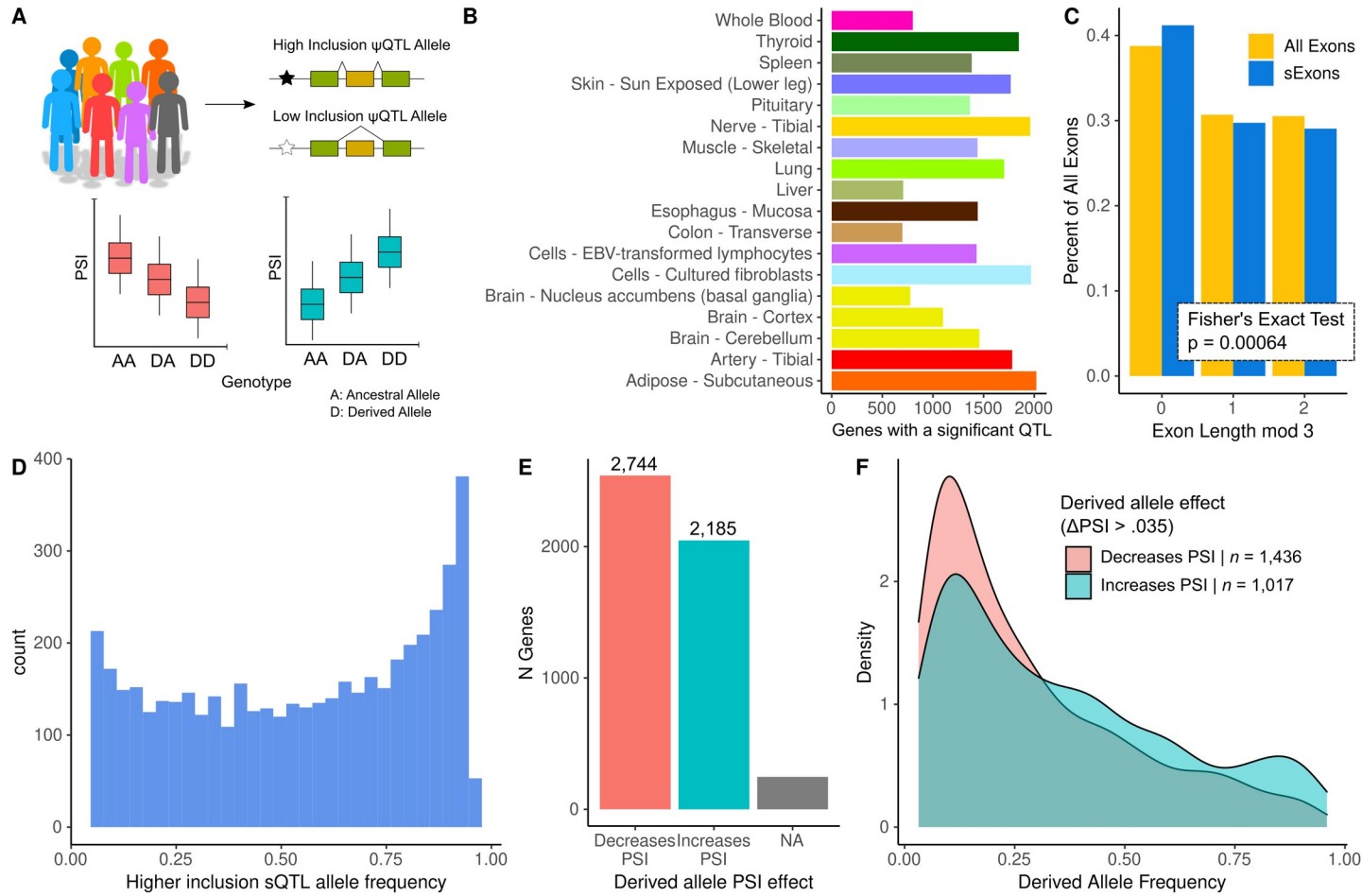

**Fig 1. Properties of genetically regulated exon splicing, as revealed by ψQTL analysis.** A) Overview of the analysis approach. Using bulk RNA-seq data from GTEx V8, we mapped splicing quantitative trait loci using individual exon PSI as a molecular phenotype. This allowed us to define a 'high inclusion' and 'low inclusion' allele, as well as define whether the ψQTL derived allele results in higher or lower exon inclusion in the final transcript. B) Number of mapped ψQTLs per GTEx tissue. We chose these 18 tissues based on their coverage of protein coding genes in GTEx. C) Percentage of symmetric exons in sExons and all exons annotated in gencode v26. We found that sExons are more likely to be symmetric, indicating that ψQTLs are less likely to induce large functional changes in target proteins. D) Distribution of high-inclusion ψQTL allele frequencies across 4,835 genes. We found that alleles which correspond to high exon inclusion are more common in the GTEx V8 dataset. E) Counts of derived allele effect directions. It was more common for ψQTL derived alleles to decrease target exon PSI. F) Distribution of derived allele frequencies between ψQTLs where the derived allele increases vs. decreases PSI. Here we limited to ψQTLs where the ΔPSI score is greater than 0.035, but this difference increases at more stringent cutoffs (see S1D Fig).

We next asked if ψQTLs are more likely to act by splicing out typically highly included exons, or splicing in typically lowly included exons. We found that across all sVariants, the major allele more often corresponded to higher exon inclusion, and that ψQTL derived alleles more commonly trigger exon skipping (Fig 1D and 1E, Binomial $p = 3.21 \times 10^{-13}$). Interestingly, we found that these derived alleles were also less common in the population, indicating potential selective pressure against loss of an exon in transcripts (Fig 1F). This effect was more pronounced when limiting to higher effect size ψQTLs (S1D Fig). While molecular QTLs are typically thought of as having little impact on fitness due to their wide distribution in the population, these results indicate that ψQTL may undergo purifying selection driven by downstream molecular effects.

Though not the main focus of this analysis, we annotated the sVariants themselves using VEP [33] to ask if derived alleles triggering exon skipping are more likely to fall in exonic,

intronic, or intergenic space with respect to their target gene. We found no significant differences in this regard, with approximately equal proportions of variants falling in each annotation category (S1E Fig).

## ψQTLs share signals between GWAS loci and eQTLs

Next, we sought to investigate if higher interpretability of ψQTLs could potentially build mechanistic insight of genome wide association study (GWAS) hits and expression QTLs.

To this end, we performed colocalization analysis [34] of ψQTLs mapped across 18 GTEx tissues against curated sets of GWAS summary statistics for 87 traits [35] (See Methods for details). Colocalization is a statistical framework to determine a posterior probability that two genetic association studies share an underlying causal variant. Out of 82,729 splicing events with a ψQTL, we found that 942 (1.13%) colocalized with at least one GWAS trait, corresponding to 338 genes out of 4,725 ψQTLs (7.15%). At least one colocalizing ψQTL in at least one tissue was found for 70 out of 87 tested traits, some replicating across multiple tissues (S3 Fig). These percentages are slightly higher than previous reports of sQTL trait colocalization (5% of genes in Barbeira et al. [35]), suggesting that ψQTLs may be prioritizing more biologically relevant splicing signals. Among genes with a colocalization event, we found no significant association with exon symmetry or derived allele effect directions, in comparison to other genes with a ψQTL but no colocalization (Fig 2B). However, we recognize that colocalization analysis is often conservative [35], and we are likely missing some splicing-trait associations that we may be underpowered to detect given the size of the dataset and the number of GWAS summary statistics available.

While it is known that eQTLs and sQTLs generally have little overlap between causal variants,[9, 12, 26] we aimed to assess if ψQTLs could reveal the mechanisms of cases where an overlap is found. One model for why this may occur is that a ψQTL triggering splicing of a non-symmetric exon induces nonsense mediated decay [32], thereby resulting in a reduction of transcript levels which is then manifested as an eQTL. To test this hypothesis, we performed fine mapping of ψQTLs using Susie [36] to prioritize potential causal variants and check for overlaps with eQTLs (Methods). Across genes and tissues, we found that ψQTLs had 1.49 credible sets per gene on average, with each credible set containing a mean of 2.19 variants. We then overlapped these credible sets with those of GTEx eQTLs on a tissue by tissue basis. While the signal was weak, we found a consistent pattern of ψQTLs with non-symmetric sExons more likely to overlap with eQTLs, in comparison to symmetric exons (Fig 2B). This highlights the possibility that genetic effects on common splicing of non-symmetric exons that disrupt open reading frames could be another mechanism for genetic effects on gene expression.

## The effects of genetically controlled splicing on protein structure

Another advantage of our ψQTL method is that it allows for straightforward mapping of mRNA splicing changes onto downstream protein structure. Since PSI is interpretable in this way, we sought to ask if exons influenced by regulatory variants have any distinguishing properties with respect to their corresponding protein domains, compared to variable exons with no significant genetic splicing regulators. We hypothesized that sExons would be depleted for highly structured protein domains since these would likely have a larger impact on protein function which might be under purifying selection. Utilizing protein structure databases, as well as the more sophisticated protein structure prediction tool Alphafold2, we built a holistic approach to probe these questions.

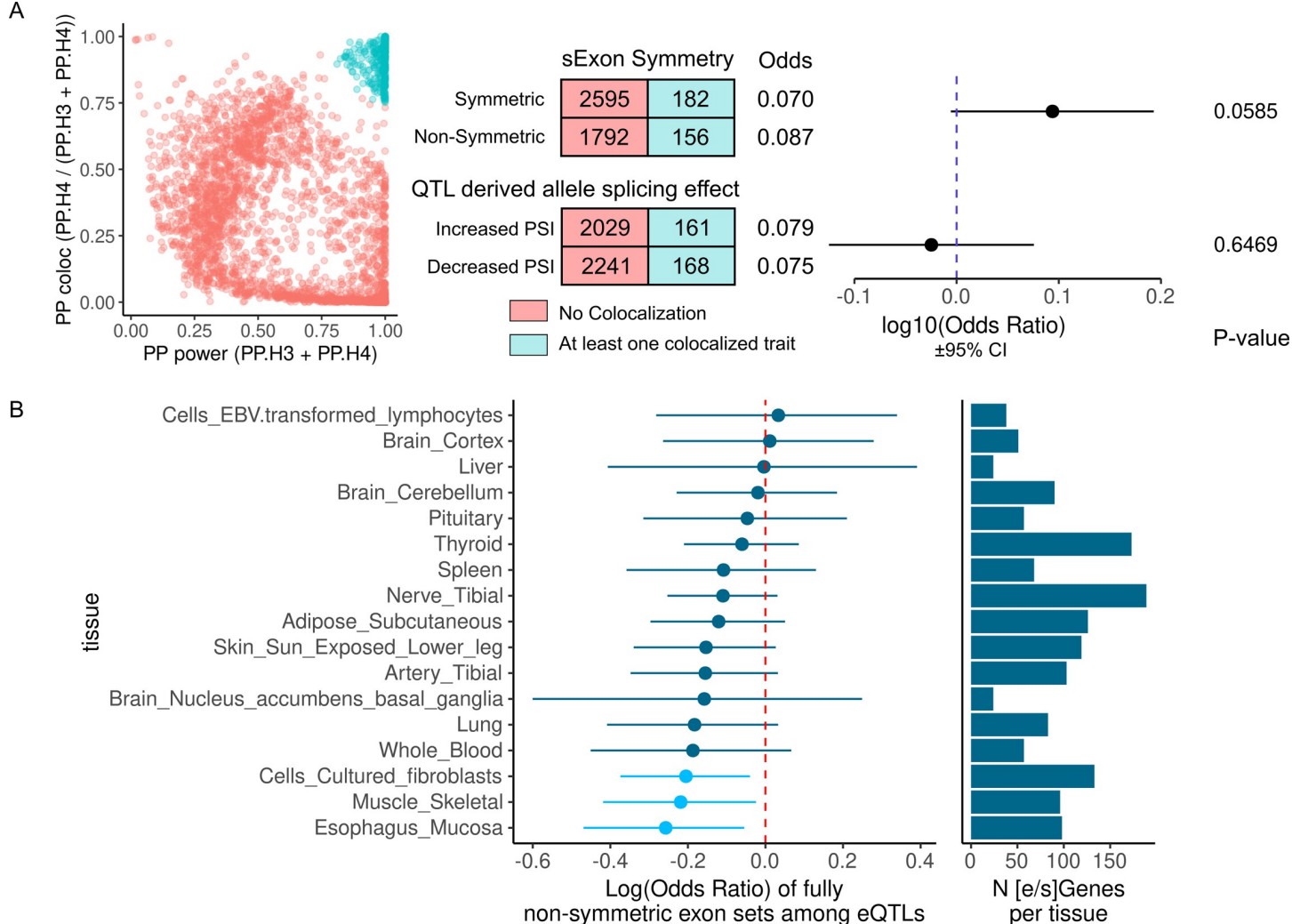

**Fig 2. ψQTL-GWAS colocalizations and shared causal variants with eQTLs.** A) ψQTL-GWAS posterior distributions for power vs. colocalization, for the top colocalization event for each ψQTL gene. A cluster appears in the upper right corner, which we use as a cutoff to define trait colocalization events (blue). Contingency tables represent the number of genes with or without a colocalization event, and where the total exon length is symmetric or nonsymmetric and where the ψQTL derived allele increases or decreases exon inclusion. Using Fisher's exact test, neither comparison reaches statistical significance. B) Log(Odds Ratio) for enrichment of non-symmetric sExons with ψQTLs that share causal variant credible sets with eQTLs, on a tissue by tissue basis. Light blue colors indicate a nominally significant enrichment (Fisher's exact test p < .05. The barplot reports the number of genes with shared eQTL and ψQTL credible set in each tissue.

To begin we mapped 4,566 non-terminal sExon nucleic acid sequences onto human protein amino acid sequences extracted from the MANE database [37] (see Methods for details). MANE isoforms represent the most commonly used protein isoforms across many public datasets, which are often the most clinically relevant with respect to variant interpretation [25]. MANE was chosen as a reference because it includes maps of nucleic acid to amino acid sequences for each gene, thus eliminating potential ambiguity in choosing the correct open reading frame. Across sExons, 2,824 (61.85%) were represented in their respective gene's MANE isoform. Notably, we found that exons with higher median PSI in GTEx were more likely to be included in MANE (S3 Fig), likely because low PSI exons are typically not a part of the most common gene isoform. We also extracted amino acid sequences from 2,708 constitutive exons and 2,071 variable exons from genes with no significant ψQTL.

Next, these pools of exons were divided and compared accordingly, to evaluate their associations with various protein features: constitutive and variable exons, variable exons with high (>0.5) and low (<0.5) median PSI, sExons and exons without a significant $\psi$QTL, and sExons that colocalize and do not colocalize with a GWAS trait. Across each pair of exon sets, amino acid sequences were annotated with multiple features describing the structuredness, solubility, length, and function of the corresponding protein domain (Table 1). The means between features were then compared using the non-parametric Mann-Whitney U-test (Fig 3).

We first focused on pLDDT and RSA scores that in combination could serve as a proxy for an exon being at the interior or exterior of the protein 3 dimensional structure [22, 38]. Overall, we observed that high PSI exons in general had higher per exon pLDDT scores ($p = 6.5610^{-39}$) and lower per exon RSA ($p = 1.1410^{-30}$). This indicates, unsurprisingly, that high PSI exons are enriched in well-structured core regions of their protein's 3D structure. Interestingly, we observed that asparagines, which are indicative of protein phosphorylation sites and functional relevance, were depleted in high PSI exons compared to low PSI exons ($p = 0.023$). We also observed depletion of asparagines in sExons when compared to variably spliced exons not regulated by any $\psi$QTL. sExons are also depleted for cysteines ($p = 0.0023$) which could indicate that alternatively spliced exons are less important for the overall protein 3D structure.

Finally we focused on sExon targets of $\psi$QTLs that colocalize with one of the 87 GWAS traits discussed in the previous section, to assess if exons whose genetically controlled splicing is involved in a trait share any discernible characteristics. The analysis revealed that these sExons overall appear to be less structured ($p = 6.1 \times 10^{-3}$ and $p = 0.034$ for RSA and pLDDT scores respectively). Additionally, colocalizing exons were enriched for asparagines and depleted for cysteines ($p = 0.023$ and $p = 0.071$ respectively), which could indicate that GWAS-relevant sExons are more likely to carry phosphorylation chemical modifications (since asparagines are frequently the residues to be phosphorylated) than those not without GWAS indication.

For further functional annotation of sExons, we analyzed the binary features including involvement in the transmembrane domain, exon symmetry, and numerous domain and

**Table 1. Description of features used to describe protein domains of interest.**

| Feature | Description and Significance |
|---|---|
| RSA | The relative solvent accessible surface area (rASA) of a residue is a degree of residue solvent exposure. RSA < 25 is considered to be buried in the protein, otherwise, it is considered to be exposed. For each sExon, we report the first quantile of RSA across all residues, which describes the overall accessible area of the exon. |
| pLDDT | Predicted local distance test. This is a per-residue metric output by AlphaFold2, which represents the model's confidence in its prediction of protein structure at that residue. A lower pLDDT score indicates an intrinsically unstructured protein domain, and vice versa. For each sExon, we report the third quartile of pLDDT scores across all residues, which describes the overall structuredness of this domain. |
| % Asparagine Residues | Percent of asparagines in the aligned sExon sequence. This metric is used because asparagines are important sites for protein phosphorylation. It could indicate functional importance of the domain of interest. |
| % Cysteine Residues | Percent of cysteines in the aligned sExon sequence. This metric is used because cysteines shape the overall 3D structure by forming disulfide bridges. |
| Length | Length of amino acid sequence aligned to the MANE Ensembl database sequences. Alternative splicing of longer exons could potentially have a higher impact on protein function. |
| Presence of Functional Domain | Percent of sExons that carry any functional domain signal. Those signals include a wide range of amino acid sequence motifs and chemical modifications as well as cellular localisation signals extracted from the UniProt database. This metric could indicate the overall importance of sExons for proteins' function. |

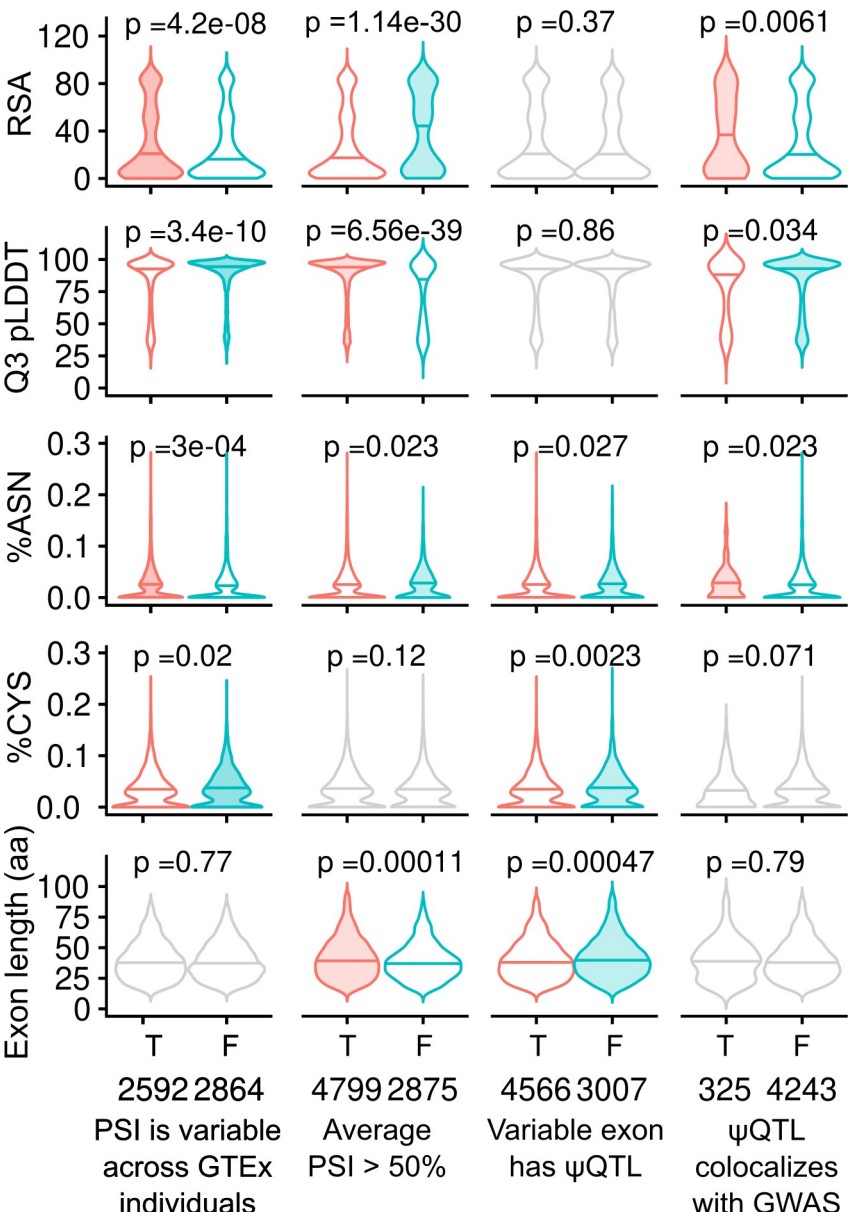

**Fig 3. Comparisons of encoded protein domain features among variably and constitutively spliced exons.** Five continuous features (see Table 1) of exons are tested across four comparisons. For each comparison, the violin representing the group with the higher median value for each feature is shaded. P-values are calculated using Mann-Whitney U-tests. Comparisons that did not reach statistical significance are plotted in grey.

motif annotations from the UniProt [21]. While no significant differences were observed in most comparisons, high PSI exons were depleted for being involved in transmembrane domains (Fisher's Exact Test p = 2.04 $10^{-42}$) and were enriched for overlapping any domain annotation signal (Fisher's Exact Test p = 3.88 $10^{-213}$). This is consistent with prior observations of higher per exon pLDDT score and lower RSA for those exons, per the previous results. The difficulty of detecting signals may be at least partially due to the incompleteness of functional domain databases, with only 2,067 out of 4,835 (42.8%) of tested exons having any

functional domain annotation. While some proteins are well-studied and annotated, the majority are still uncurated.

## Genetically controlled protein structural changes in GWAS trait colocalization

Next, we sought to evaluate if specific changes in protein structure were attributed to ψQTLs with strong GWAS trait colocalization signals. To do so, we predicted the structure of 146 proteins for both spliced-in and spliced-out sExon isoforms with ColabFold [39], quantifying which regions of the protein are affected by alternate exon usage and to what degree. The chosen exons have a ψQTL that colocalizes with a GWAS trait, are non-terminal, and both isoforms' length is less than ∼1200 amino acids, due to the current limitations of ColabFold. Overall, we observed a wide range of structural rearrangements, from minor deletions of unstructured regions to the exclusion of whole structured domains. To summarize observed perturbations across genes, Euclidean distances were calculated between predicted alignment error (PAE) [22] score matrices for each pair of structures. Cases where splicing causes major rearrangements on the structural level are expected to have a higher Euclidean distance between the two isoforms. In general, we found that a wide range of structural changes was driven by alternative splicing associated with GWAS traits (Fig 4). We find that this Euclidean distance between pairs of isoforms correlates weakly with the gene's LOEUF score ($\rho = 0.205$, $p = 0.013$, S3A Fig), indicating that more constrained genes with respect to loss of function variant intolerance are also less tolerant to large structural changes. As a measure of goodness of structural alignment, root mean square distance (RMSD) was calculated between spliced-in and out isoforms as well (S4A Fig). This indicates the quality of alignment by showing the mean distance (in Å) between corresponding residues in structurally aligned proteins. RMSD could serve as a proxy for topological rearrangement on the 3D level. In contrast, the Euclidean distance between PAE matrices indicates how well domains between structures are preserved, and in general correlated poorly with the Euclidean distance score (S4B Fig). All 146 predicted structures are available to download as supplementary data (https://zenodo.org/records/8137337).

Finally, we focus on three notable examples of predicted protein structure changes associated with GWAS hits, which highlight the utility of our approach.

We first investigated structural perturbations of SP140 caused by the splicing of exon 13 (Fig 5A). It is associated to chr2:230245867:C:T (rs28445040), which is also colocalized with a Crohn's disease GWAS. This result was first reported in Zhao et. al. [40], and we replicated their findings using entirely different datasets. (In Zhao et. al, exon 7 is the significant exon defined by Ensembl v65. This maps to exon 13 in gencode v26, which we use here.) The sExon is aligned to amino acids 223 to 247 of MANE transcript of SP140. No structural annotation is available for this part of the protein, as currently available experimental structures only cover positions 687–862 (pdb accession: 6g8r). Structural changes predicted by AlphaFold2 between 2 isoforms are minor and are in an intrinsically disordered region of the protein that is supported by a relatively small euclidean distance (1.73) although RMSD is quite high (16.9). Our predictive approach corroborates exon 13 lying within an unstructured region, and that its alternative splicing does not affect the protein core. No known post-translational modifications or protein-protein interaction annotations in structural databases overlap with the peptide.

Next, we present another case of a ψQTL with a significant GWAS colocalization event: chr11:584591:A:G (rs35865896) associated with splicing of exon 4 in IRF7, a key transcription factor of the immune system. The QTL is significant in 12 out of the 18 tested tissues. This association implies that higher usage of an isoform that skips exon 4 is associated with

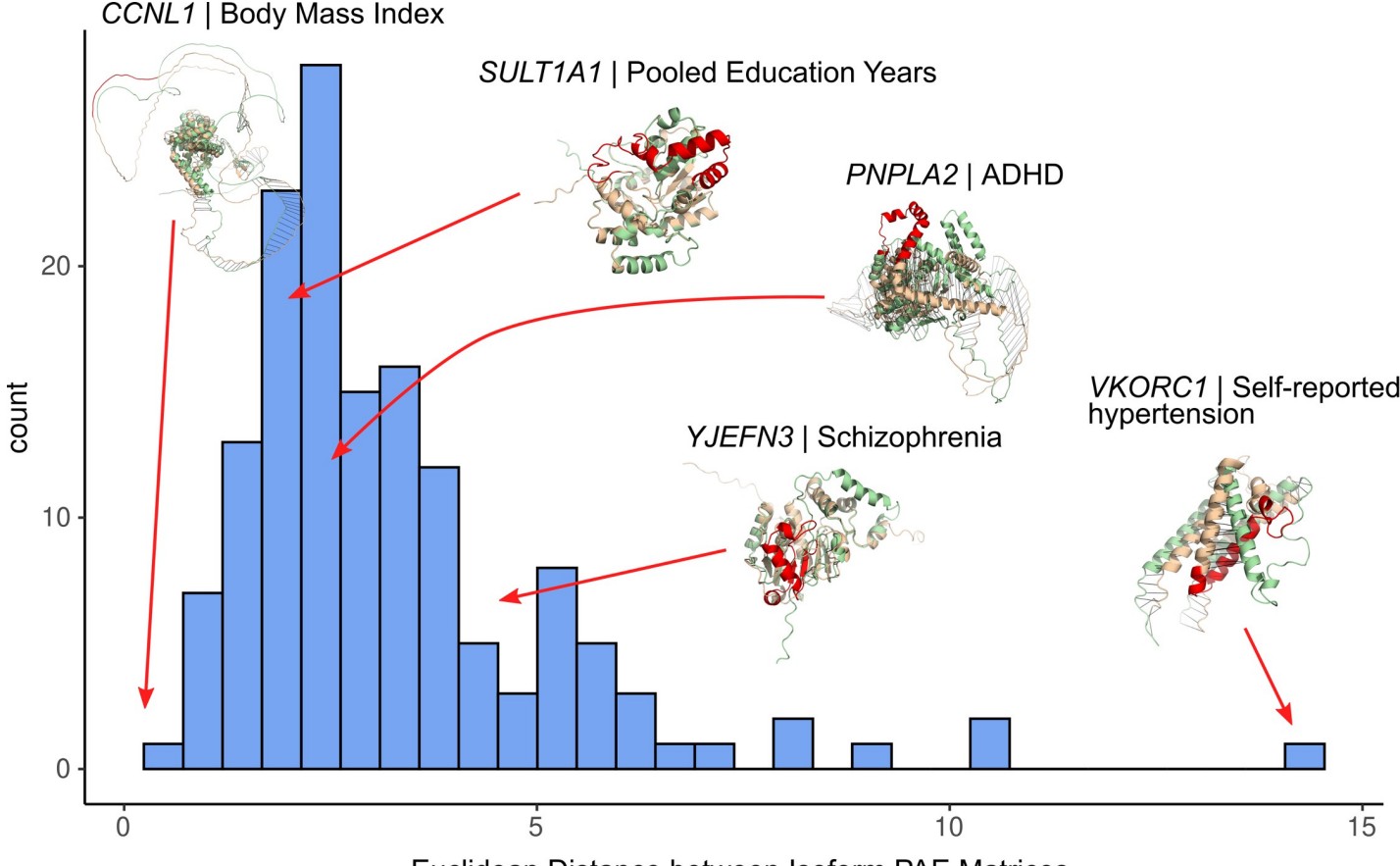

**Fig 4. Distribution of Euclidean distances between predicted alignment error (PAE) matrices among GWAS colocalized ψQTLs.** A higher Euclidean distance indicates more structural difference between the spliced in and spliced out isoform. Seven notable examples are showcased, with the spliced in and spliced out isoforms overlaid. The tan structure is the spliced in, and green structure is the spliced-out isoform. The red region of the protein represents the sExon. Each protein is labeled by its gene and the GWAS trait it is associated with.

decreased risk of lupus erythematosus [41] (Fig 5B). The exon aligns to positions 158–226 in the canonical transcript from MANE, and we observed modest structural changes (Euclidean distance 2.46) between the canonical and spliced-out predicted structures (RMSD = 17.4). After aligning the two structures, we found that the whole C-terminal domain of IRF7 is mirrored. However, the overall organization of the domain is preserved. Notably, cleavage sites of 2 viral proteases 3C (positions 167–168 and 189–190 for EV68 and EV71 respectively) are present in the sExon [42]. In addition, 2 residues of the alternatively spliced exon are involved in DNA binding (amino acids 187 and 189) [43]. It could potentially be of interest to investigate the effect of splicing on the DNA binding abilities of IRF7 with respect to lupus risk and progression, as suggested by this finding.

Finally, we focus on alternative splicing at exon 16 of HMGCR, which is part of a cholesterol metabolism pathway (Fig 5C). We cataloged chr5:75355259:A:G (rs3846662) as the top sVariant, which strongly colocalizes with a GWAS for high cholesterol levels. We observed a substantial Euclidean distance between the 2 predicted structures (3.14 compared to the median of 2.64 across all other comparisons). While, according to the PAE matrix, the spliced-in isoform contains 2 major clusters (domains), the middle part of the C-terminal domain in the spliced-out isoform is predicted to break into 2 domains. According to the prediction

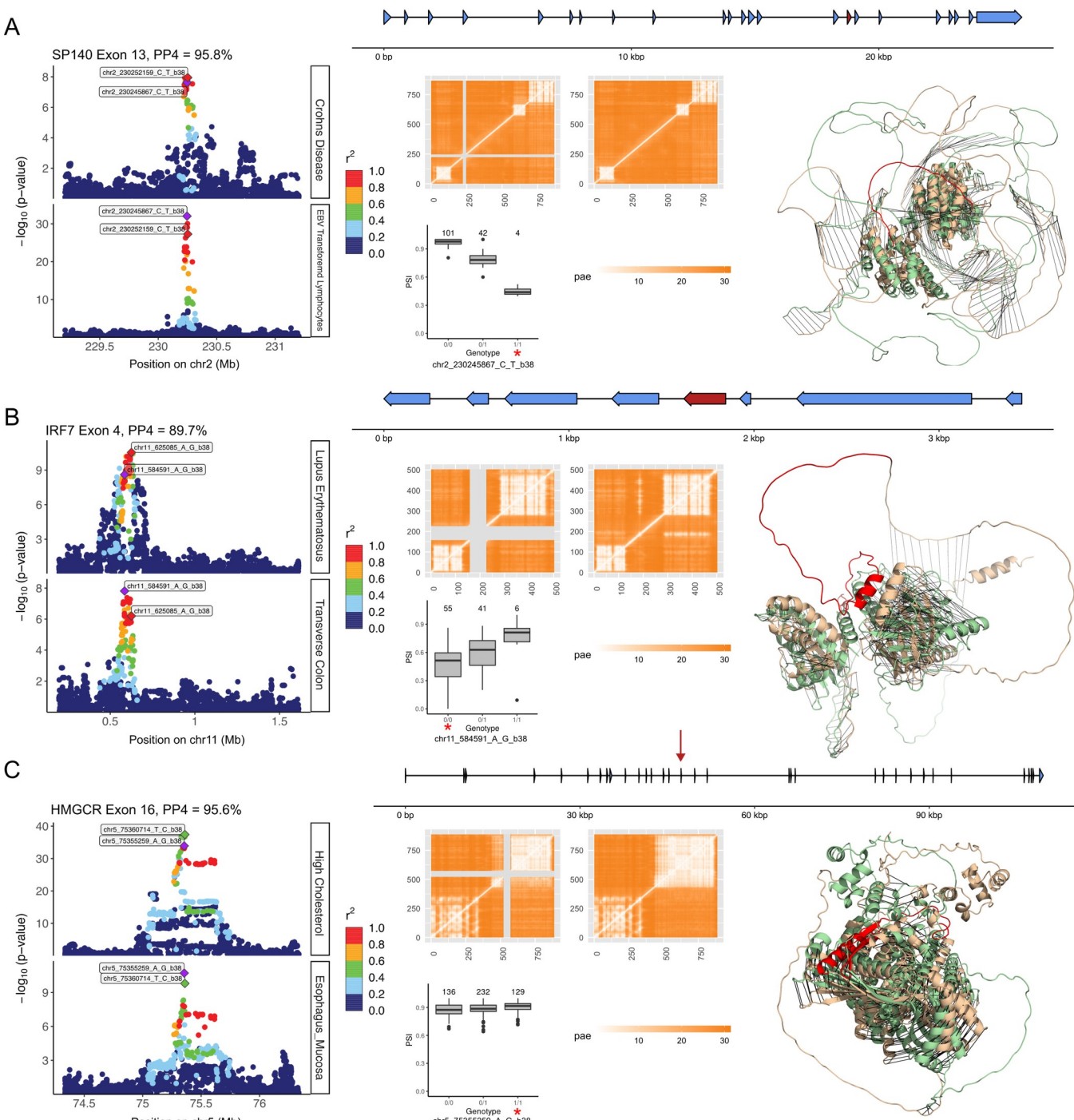

**Fig 5. Predicted structural changes associated with GWAS-colocalized ψQTLs.** Detailed descriptions of splicing of A) SP140 exon 13, colocalized with Crohn's disease risk, B) IRF7 exon 4, colocalized with Lupus Erythematosus risk and C) HMGCR exon 16, colocalized with high cholesterol levels. The whole gene is displayed at the top of each row, with the relevant exon labelled in red or with a red arrow. In each row, from left to right, we display: Two overlapping locuszoom plots for the GWAS (top) and the ψQTL (bottom) in the same region. Variants are colored by their LD with respect to the top variant for each association, measured in $r^2$. To the lower middle, the plot shows the ψQTL in each respective GTEx tissue, with the genotype corresponding to reference and alternate alleles. The GWAS risk allele is marked with a red asterisk. Next, we display two matrices of PAE scores, corresponding to the AlphaFold2 predicted structures of spliced-in and spliced-out exons. This is a simple way to visualize splicing a highly structured protein domain. The sExon is marked in grey in the spliced-out isoform matrix. Third, we plot ribbon diagrams with overlapping structures of the two isoforms, as predicted by AlphaFold2. The tan structure is the spliced-in, and the green structure is the spliced-out isoform. The variably spliced exon is coloured in red.

followed by structural alignment (RMSD = 6.13), splicing of exon 16 that corresponds to amino acids 522–574 in the MANE Ensembl protein database isoform causes mirroring of a part of the protein between amino acids 368 and 511, and separation to a new structural unit in the spliced-out isoform. Follow-up sequence annotation revealed that the exon consists of a turn (522–525), 2 beta strands (528–546 and 549–556 respectively) and a helix (562–575) inferred from the PDB structural database (PDB accession: 2r4f). In addition, this exon is annotated with one of three Coenzyme A binding domains found in HMGCR (565–571) [44]. Interestingly, HMGCR is responsible for a rate-limiting step in the synthesis of cholesterol, thus regulating cellular cholesterol homeostasis. Taking this into account, we hypothesize that skipping exon 16 in HMGCR interferes with the Coenzyme A binding domain, thus inhibiting this enzyme's function and reducing blood cholesterol levels. Even though the $\psi$QTL effect size is quite small in this example ($\Delta$PSI = 0.014), the GWAS risk allele and the exon-including $\psi$QTL allele are the same, further suggesting a functional relationship.

In general, we present a powerful approach for characterizing splicing-related protein changes. While identifying distinct protein isoforms originating from mRNA splicing has been challenging historically [45], computational prediction provides a path for identifying structures of minor isoforms whose usage may be important for trait risk. Importantly, our technique relates dosages of isoforms that include or skip exons to common genetic variation. Predicting these structural changes as related to genetics adds another layer of interpretation when deciphering the mechanisms that link genetic variants to GWAS traits, and in the future could assist in identifying drug targets for genetic diseases.

## Discussion

In conclusion, we present a perspective to splice quantitative trait loci mapping that prioritizes downstream interpretability of the splicing signal itself. In particular, we focus on the properties of $\psi$QTL-affected exons, and the impact of genetically controlled slicing on protein structure. These aspects of splicing are generally understudied, as most research to date has focused on characterizing the properties of splicing variants themselves, as opposed to the molecular consequences of alternative splicing [9, 15, 18, 26]. Through our approach of focusing on exon skipping events, we found that symmetric exons are more likely to be affected by $\psi$QTLs, and that derived alleles are more likely to trigger exon exclusion. We found many instances of $\psi$QTLs that colocalized with GWAS traits, and that these trait-relevant splicing events were more likely to occur in the core structured regions of proteins. By predicting protein structures with AlphaFold, we demonstrate the potential mechanisms of different changes in splicing leading to trait associations.

When calculating PSI using counts of reads that span exon-exon junctions, we depend solely on exon annotations, rather than whole isoform annotations. This is advantageous in our case, as isoform annotations are notoriously incomplete [46, 47], and methods to estimate ratios of isoforms can be unreliable [48, 49]. However, we recognize that exon skipping is far from the full picture of splicing variation, and hypothetical structures with and without a single exon may not be translated in reality. Importantly, it should be noted that our method is not meant to predict ratios of real transcripts or protein isoforms, but rather to prioritize domains that may have relevance for fitness and trait associations. With newer advanced methods like long read sequencing, it is becoming possible to quantify whole transcripts which capture complex splicing events [50–52]. In the future, these technologies could provide a higher resolution picture of isoform proportions associated with trait and disease risk. Additionally, while in this work we focus on splicing changes associated with common variants, resolving

structures of rare splice isoforms is a fruitful approach for improving genetic diagnosis, discovering rare disease etiology, and identifying potential therapeutic targets [53–56].

With the revolution in protein-structure prediction launched by AlphaFold [22], it is now possible to predict isoform structures associated with conditions of interest, thus opening up enormous opportunities to track molecular perturbations without performing laborious structural biology experiments. However, this method comes with some limitations. As such, co-translational and post-translational modifications are not predicted by AlphaFold, while these potentially drive a large fraction of cell-signaling and trait associations [57, 58]. Additionally, AlphaFold remains limited to single protein structures with a length limitation of 2700 residues (1000–1200 residues in the case of ColabFold used in this study), although recent model upgrades for predicting protein complexes were released by DeepMind [59]. Another area of interest in interpreting AlphaFold predictions is determination of intrinsically disordered or unstructured protein regions, as low pLDDT regions are thought to have a high likelihood of being unstructured in isolation. It has been argued that AlphaFold may be of use as a tool for identifying such regions, performing on par with specifically created tools. However, experimental validation of intrinsically disordered regions is highly recommended using SAXS, NMR, X-Ray crystallography, cryo-EM, etc. Additionally, AlphaFold has not been trained or validated for predicting the structural impact of single mutations, and generally performs poorly for this purpose [60]. Here, we focus on the more tractable problem of comparing isoform structures where large portions of the protein differ, rather than single mutations, where the model's output is likely more biologically relevant. Although structures obtained from AlphaFold2 generally correlate well with experimental structures, regions with low confidence scores should be treated cautiously as they might not represent truly disordered regions [61]. Thus, we reiterate that predicted structures do not necessarily correspond to actual biological structures, but rather prioritize protein domains that may be affected by genetically regulated splicing.

While determining the structure of a protein gives valuable information about its function and role in specific conditions, the structure is not the only relevant factor. Post-translational modifications and cellular localization play a crucial role in protein activation and deactivation [62, 63]. To address how those properties are changed in alternatively spliced transcripts, we utilized the UniProtKB database which is a well-curated source of various features' annotation. Although it is a trustworthy and fast-expanding resource, functional annotations for many proteins are still incomplete. Mostly those are proteins not involved in disorders, common pathways, and other well-studied processes. This limits UniProtKB's utility for discovering new associations between structural changes and molecular/functional perturbations. Additionally, disordered regions are in general poorly annotated as they are not well captured by standard protein structure determination methods [64]. To fill the gap of experimental annotation, multiple prediction tools have been developed [43]. While it is beneficial to have at least some annotation, one should treat it with caution.

Despite limitations, ψQTL analysis provides a different perspective on genetically controlled pre-mRNA splicing. Our findings indicate that the effect size and direction of exon skipping events affect variant allele frequencies, which implies an association with overall fitness. Paired with the computational prediction of protein structures, we envision this technique being used as a starting point for contextualizing genetic associations to disease where alternative splicing is suspected to be involved. In future studies of splicing QTLs, we suggest that the impact on protein structure be considered further.

## Methods

### PSI calling from GTEx V8

Exon level percent spliced in (PSI) scores were calculated from GTEx V8 RNA-seq BAM files (See Consortium 2020 [26] Supplemental Information for upstream data processing steps). We limited our analysis to 18 tissues, which were chosen for their coverage in GTEx and their coverage of the most coding genes possible (S1 Table). Exon PSI for protein-coding genes was quantified using the Integrative Pipeline for Splicing Analysis (IPSA), [65, 66] which was modified to run on Google Cloud through Terra. (https://github.com/guigolab/ipsa-nf) The '-unstranded' flag was used during the sjcount process. Exons were defined by the modified version of Gencode annotation v26 used in GTEx V8, which collapses genes with multiple isoforms to a single isoform per gene. (https://storage.googleapis.com/gtex_analysis_v8/reference/gencode.v26.GRCh38.genes.gtf).

For downstream QTL analysis, PSI data for each tissue was prepared by 1) removing exons with data available in less than 50% of donors and 2) removing exons with fewer than 10 unique values across all available donors, to remove constitutive exons with no variability across individuals (S1 Table). Overall, we kept between 9.41% and 17.16% of exons with PSI data available. In subsequent analyses, this set of exons is referred to as "sufficiently variable." Post-filtering exon PSI calls were normalized for QTL mapping by randomly breaking any ties between two individuals with the same PSI at an exon, then applying inverse-normal transformation across all individuals. Filtered and normalized PSI calls were saved in BED format with start/end position corresponding to each gene's transcription start site (TSS). The gene containing each exon was included in the BED files for use with QTLtools' group permutation mode.

Additionally, constitutive exons were separated from variable exons for other downstream analyses. These were defined by 1) selecting all exons with a PSI of 1 in in all but at most 10 donors across the 18 GTEx tissues 2) Merging this list across all 18 tissues, recording the number of times an exon is constitutive across tissues 3) Keeping exons that were constitutive across at least 9 tissues 3) Further filtering the list by removing terminal exons, and limiting to one constitutive exon per gene, based on *a*. the number of tissues an exon was constitutive in and *b*. a random selection in the few cases where there were ties.

### Primary ψQTL mapping and collapsing across tissues

For each of the 18 GTEx V8 tissue groups, *cis*-QTL mapping was run on every exon that passed filtering, considering all genetic variants with an allele frequency greater than 5% in GTEx within 1Mb of the gene's TSS. We used QTLtools [67] run in grouped permutation mode, with groups defined by gene. This strategy controls for splicing correlation between exons that are part of the same gene. 15 PEER factors, 5 genetic principal components (PCs), as well as sex, PCR bias, and sequencing platform were also included as covariates in the QTL model, as recommended in the GTEx V8 STAR methods [26].

For every exon, we selected the most significant variant, and for every gene the most significant exon. A gene was determined to be a ψQTL if the top variant's beta adjusted p-value was less than 0.05. Although QTLs were directly mapped using normalized PSI measurements, we defined effect sizes by referring back to the non-normalized PSI calls and calculating the change in PSI (ΔPSI) as difference in the REF/REF and ALT/ALT genotype medians.

We compiled the ψQTL results across tissues to achieve a set of cross-tissue top ψQTLs. When a gene was significant across multiple tissues, we used the tissue where the effect size (ΔPSI score) of the ψQTL was the highest. This process ensured that a gene was only included

once in our final set of ψQTLs, and was labeled by one variant that affects splicing (sVariant). Underlying LD patterns may obscure the true variant that causes splicing differences, but for simplicity in this project, we choose a single sVariant per exon.

To evaluate concordance of ψQTLs across GTEx tissues, we first ran a nominal QTL calling pass on all tissues, then extracted the p-values of significant sVariant-sGene pairs in a discovery tissue from all 17 other test tissues. We used the Storey and Tibshirani q-value approach to evaluate the fraction of true positives ($\pi_1$) discovered in each test tissue.

For each cross-tissue top ψQTL, we labeled the alleles associated with high and low target exon inclusion based on the regression slope from QTL calling. This classification is more biologically relevant than reference and alternative alleles, which are only dependent on the reference genome. Additionally, we labeled the ancestral and derived alleles of each top ψQTL based on data from the 1000 Genomes Project Phase 3 [68].

To compare sExons to variable exons without a ψQTL, we considered genes where the most significant variant-exon pair across all tissues in which the gene was tested had an adjusted p-value > 0.2.

**ψQTL mapping in Geuvadis.** To check with concordance between GTEx ψQTLs and another dataset, we performed the same procedure for PSI calling and QTL calling in Geuvadis [30] as for GTEx. Since Geuvadis RNA sequencing and genotyping are aligned to hg37, we first used liftover to convert the coordinates for gencode v26 exons from hg38 to hg37. We then ran our pipeline as with GTEx. In total, we used 462 individuals from Geuvadis with European ancestry, and performed QTL calling using 3 genetic principal components as covariates, in addition to PEER factors and sequencing center. To compare Geuvadis sVariants with those in GTEx, we converted coordinates back to hg38.

## Colocalization analyses

We performed colocalization analysis to evaluate the extent that ψQTLs share potential causal variants with GWAS traits. First, we ran a nominal QTLtools pass in *cis* using PSI calls from exons with a significant ψQTL in at least 1 of 18 GTEx tissues as in the previous analysis. The definition of a common variant and range of 1Mb up and downstream of the gene's TSS were the same. With this set of common variant-splicing associations, we performed Approximate Bayes Factor colocalization analysis using the coloc R package [34], running nominal ψQTL calls from 18 GTEx tissues against 87 sets of GWAS summary statistics (refer to Barbeira et al. [35] for the full list) for a total of 1,566 possible colocalization events. To define a colocalized trait, we calculated PP.power and PP.coloc for each potential colocalization event, which we define as (PP.H3 + PP.H4) and (PP.H4 / (PP.H3 + PP.H4)) respectively. We considered a trait to be colocalized if the Euclidean distance between (1,1) and (PP.power, PP.coloc) is less than .25 (See Fig 2). We chose this looser definition of colocalization to allow for more data in downstream analyses, and where the false positive rate is less critical.

## Fine mapping with Susie (overlap with eQTLs)

To find causal variant credible sets for ψQTLs, we applied the fine mapping procedure used in the eQTL catalog [69], which applies Susie to find independent sets of variants with 95% posterior inclusion probability of containing the true causal variant for a QTL. (https://github.com/eQTL-Catalogue/qtlmap) We ran fine mapping on all exons independently, so we employed an aggregation procedure to achieve one or multiple sVariant credible sets per gene. For all exons in a gene, we considered all possible unions of all credible sets. Collapsed credible sets contained disjoint sets of variants. We then used these collapsed credible sets to compare to eQTL credible sets, also from the eQTL catalog.

### Extraction of amino acid sequences and sequence properties

To analyze how exons affected by ψQTLs map to protein sequence and structure, we leveraged AlphaFold predictions and other resources. First, sExon nucleic acid sequences were extracted from the hg38 assembly of the human genome using gencode.v26.GRCh38.GTExV8.genes annotations. blastx: 2.12.0+ was used to perform mapping of extracted sExons onto transcript sequences present in the MANE.GRCh38.v1.0 database. For each sExon best hits were selected based on e-value. Hits with e-value greater than 0.001 were discarded. sExons with the best hit occurring in another protein were also excluded from further analysis.

Length outliers were removed before conducting structural analysis. pLDDT scores were obtained from pdb files of the human proteome from the AlphaFold database (Reference proteome UP000005640). As the pLDDT score is a per residue measure, summary statistics (min, Q1, median, Q3, max) were calculated for each exon. (https://freesasa.github.io/) FreeSASA 2.0.3 (—format = rsa) tool was used to calculate per residue relative solvent accessibility (RSA). The same summary statistics were applied as for the pLDDT score. Domain annotation (evidence of overlap between sExon and any annotated domain/motif/chemical modification) was obtained from the UniProt database. The following signals were collected: DOMAIN, SIGNAL, TOPOLOGY, TRANSMEMBRANE, MOTIF, TOPO_DOM, ACT_SITE, MOD_RES, REGION, REPEAT, TRANSMEM, BINDING, NP_BIND, COILED, DISULFID, CARBOHYD, DNA_BIND, CROSSLNK, ZN_FING, METAL, SITE, INTRAMEM, LIPID for the further analysis. In addition, exons were annotated with their respective gene's LOEUF scores, which were extracted from GNOMAD v2.1.1.

To analyze distinct structural characteristics of exons, we calculated the above features across all exons with sufficient coverage in GTEx, then compared features by: (1) variable vs. constitutive exons, as defined above,(2) Highly included vs. lowly included exons, defined as sufficiently variable exons with a median PSI across all available individuals and tissues less than and greater than 50% respectively, (3) sExons vs. non-sExons, defined as exons with or without a significant ψQTL variant in at least one of the 18 analyzed GTEx tissues in the previous section, and (4) colocalizing sExons vs. non-colocalizing sExons, with the former having a significant sQTL-GWAS colocalization event signal and its absence for the former group, defined using the same cutoff thresholds as in the previous colocalization analysis. To perform statistical analysis, Mann-Whitney U-tests were performed for numerical features (RSA and pLDDT summary statistics, asparagine and cysteine percent, exon length and symmetry distributions). Fisher's exact tests were performed to test differences in categorical features (presence of the domains, enrichment in transmembrane domains, etc).

### Prediction of protein structure changes with AlphaFold2

ColabFold (https://github.com/sokrypton/ColabFold) (Google Colab version of AlphaFold 2) was used to predict protein structures for transcripts with and without sExons. To build MSA, the MMseqs2 searching tool was used on the UniRef + Environmental databases. Both paired and unpaired sequences were utilized for MSA construction. No template mode was used in order not to introduce bias of one structure by the other, since in most cases only either spliced-in or spliced-out structure is present in PDB database. Three recycles were performed to obtain better structure predictions. Two metrics were used to track potential effect of splicing event on protein structures: RMSD between aligned structures and Euclidian distance between positional alignment error (PAE) score matrices.

Prior to calculating RMSD, spliced-in and spliced-out isoforms were structurally aligned using PyMol 2.5.0. Then RMSD was calculated between aligned parts of structures, eliminating misaligned parts including alternatively spliced exons. PAE scores were analyzed to determine

perturbations in domain arrangement between spliced-in and spliced-out versions of the transcript. Euclidean distance between PAE matrices of splice-in and spliced-out isoforms was calculated to track the effect of splicing onto domain rearrangements caused by splicing events. In addition to PAE matrix analysis, structures were visualized using PyMol 2.5.0 and superimposed to detect major topological differences between them.

## Supporting information

**S1 Fig. Supplemental characterizations of ψQTLs.** A) Relationship between the number of individuals tested and the number of significant ψQTLs per tissue. We catalog more significant genes in tissues where more donors are available, as is typical in QTL studies. B) Distribution of lengths between variable exons with and without a significant sVariant controlling splicing levels. While statistically significant, the difference is not large. C) Density plot of variable exons' relative position in their respective transcripts. Across both groups, variable exons tend to occur later in the transcript. D) Increase in difference between derived allele frequency distributions when increasing the ψQTL effect size cutoff. The Kolmogorov-Smirnov D score, which quantifies the degree of difference between two distributions, increases as we consider stronger ψQTLs. E) Annotations of top sVariants across all ψQTL genes, split by derived allele effect direction on PSI. F) ψQTL replication heatmap between GTEx tissues. For a test tissue, we extract the nominal $p$-value of all significant (beta-corrected p < .05) variant-exon pairs from the 17 other test tissues, where data is available. From these sets of $p$-values, we calculate Storey's $\pi_1$ value, which represents the estimated fraction of true positive ψQTLs that were first discovered in the test tissue. The y axis is labeled with the same color key as the x axis. G) $\pi_1$ scores, calculated by retrieving the p-value from all significant sQTLs in Garrido-Martín et al. [18] and testing the same variant in the ψQTLs. H.) Percentages of tested genes with a significant ψQTL across tissues, compared to genes with an sQTL with splicing mapped using Leafcutter.
(TIF)

**S2 Fig. Tissue specificity of GWAS-ψQTL colocalization.** Counts of ψQTL-GWAS colocalization events across the 18 tested tissues. Traits are colored by their broad category, and rows are organized by hierarchical clustering.
(TIF)

**S3 Fig. Additional properties of exons found in MANE isoforms.** A) Distribution of median PSI scores between exons included or excluded in their respective gene's MANE isoform. B) Genes with larger structural changes between trait-associated isoforms are also less likely to be haploinsufficient. This suggests genes which tolerate regulatory variants with large splicing effect sizes are also more tolerant to loss-of-function coding variants.
(TIF)

**S4 Fig. RMSD between predicted protein structures.** A) Distribution of Root Mean Squared Distance (RMSD) between spliced in and spliced out ψQTL isoforms that colocalize with a GWAS trait. B) Correlation between RMSD and the Euclidean Distance between isoform PAE matrices. These two quantities are not significantly correlated.
(TIF)

**S1 Table. Total number of exons with PSI covered across tissues.**
(DOCX)

## Acknowledgments

We would like to thank the current and former members of the Lappalainen Lab for helpful discussions and code sharing.

## Author Contributions

**Conceptualization:** Jonah Einson, Tuuli Lappalainen.

**Data curation:** Mariia Minaeva.

**Formal analysis:** Jonah Einson, Mariia Minaeva, Faiza Rafi.

**Funding acquisition:** Tuuli Lappalainen.

**Investigation:** Jonah Einson, Mariia Minaeva.

**Methodology:** Tuuli Lappalainen.

**Software:** Jonah Einson.

**Supervision:** Jonah Einson, Tuuli Lappalainen.

**Visualization:** Mariia Minaeva.

**Writing – original draft:** Jonah Einson, Mariia Minaeva.

**Writing – review & editing:** Tuuli Lappalainen.

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
