## [Editor Report · Decision Letter 0]

16 Jan 2023

PONE-D-23-00371The impact of genetically controlled splicing on exon inclusion and protein structurePLOS ONE

Dear Dr. Einson,

Thank you for submitting your manuscript to PLOS ONE. After careful consideration, we feel that it has merit but does not fully meet PLOS ONE’s publication criteria as it currently stands. Therefore, we invite you to submit a revised version of the manuscript that addresses the points raised during the review process.

We look forward to receiving your revised manuscript.

Kind regards,

Chunyu Liu

Academic Editor

PLOS ONE

Journal Requirements:

"We would like to thank the current and former members of the Lappalainen Lab for helpful discussions and code sharing. This work was supported by NIH grants R01GM122924, R01MH106842, and grant WASPDDLS21:080 by the Data Driven Life Science program by the Knut and Alice Wallenberg Foundation. Part of the computations were enabled by resources provided by the Swedish National Infrastructure for Computing (SNIC) at UPPMAX partially funded by the Swedish Research Council through grant agreement no. 2018-05973. T.L. is a paid advisor to GSK, Pfizer, Goldfinch Bio and Variant Bio, and has equity in Variant Bio."

"This work was supported by the National Institutes of Health grants R01GM122924, R01MH106842 (J.E. & T.L.), the Data Driven Life Science program by the Knut and Alice Wallenberg Foundation grant WASPDDLS21:080 (M.M. & T.L.) and the S. Jay Levy Foundation (https://www.ccny.cuny.edu/sjaylevy) (F.R.). Part of the computations were enabled by resources provided by the Swedish National Infrastructure for Computing (SNIC) at UPPMAX partially funded by the Swedish Research Council through grant agreement no. 2018-05973. (M.M. & T.L.)"

"I have read the journal's policy and the authors of this manuscript have the following competing interests: T.L. is a paid advisor to GSK, Pfizer, Goldfinch Bio and Variant Bio, and has equity in Variant Bio."

Additional Editor Comments:

This is a very interesting, creative study. It is important to include some additional data and analyses to show the reproducibility of the findings. There are plenty of RNA-seq data other than GTEx, can be used.

Additional comparison across tissues is also useful. The bottom line is that we need to see that the findings, primarily those QTLs, are robust.

Once this is satisfied, we will send it out for a thorough review. Hopefully, it will be fast.

---

## [Author Response · Author response to Decision Letter 0]

13 Feb 2023

Updated Funding Statement, Competing Interest Statement, and Data Availability Statement are included in the document titled "Response to Reviewers - 2.13.23"

---

## [Decision Letter · Decision Letter 1]

17 May 2023

PONE-D-23-00371R1The impact of genetically controlled splicing on exon inclusion and protein structurePLOS ONE

Dear Dr. Einson,

Thank you for submitting your manuscript to PLOS ONE. After careful consideration, we feel that it has merit but does not fully meet PLOS ONE’s publication criteria as it currently stands. Therefore, we invite you to submit a revised version of the manuscript that addresses the points raised during the review process.

We look forward to receiving your revised manuscript.

Kind regards,

Chunyu Liu

Academic Editor

PLOS ONE

Journal Requirements:

Reviewers' comments:

Reviewer's Responses to Questions

**Comments to the Author**

1. If the authors have adequately addressed your comments raised in a previous round of review and you feel that this manuscript is now acceptable for publication, you may indicate that here to bypass the “Comments to the Author” section, enter your conflict of interest statement in the “Confidential to Editor” section, and submit your "Accept" recommendation.

Reviewer #1: (No Response)

Reviewer #2: All comments have been addressed

2. Is the manuscript technically sound, and do the data support the conclusions?

Reviewer #1: Yes

Reviewer #2: Yes

3. Has the statistical analysis been performed appropriately and rigorously? 

Reviewer #1: Yes

Reviewer #2: Yes

4. Have the authors made all data underlying the findings in their manuscript fully available?

Reviewer #1: Yes

Reviewer #2: Yes

5. Is the manuscript presented in an intelligible fashion and written in standard English?

Reviewer #1: Yes

Reviewer #2: Yes

6. Review Comments to the Author

Reviewer #1: In the Einson et al. manuscript, the authors identified splicing quantitative trait loci (sQTLs) using exon percent spliced in (PSI) scores as a quantitative phenotype. They also utilized AlphaFold2 to predict changes in protein structure associated with sQTLs that overlapped with GWAS traits, highlighting a potential new use-case for this technology in interpreting genetic effects on traits and disorders. This study presents novel insights into the effects of sQTLs on splicing events and GWAS signals.

However, there are some major comments to be addressed.

1. The study lacks a direct comparison with those protein coding genes with sQTLs from previously published studies, such as the one by Garrido-Martín et al. (2021)

Garrido-Martín, D., Borsari, B., Calvo, M. et al. Identification and analysis of splicing quantitative trait loci across multiple tissues in the human genome. Nat Commun 12, 727 (2021). https://doi.org/10.1038/s41467-020-20578-2

2. It is not clear which exon is considered the "top exon" in the statement "removing genes where the 3' or 5' terminal exon was the top exon." If the top exon refers to the first or last exon, then the majority of genes would be removed since most genes have a 3' or 5' exon.

3. While the annotations (e.g., exonic, intronic, or intergenic) of sVariants are provided, there is no information about whether an sVariant is deleterious or benign.

4. The threshold (i.e. Euclidean distance between (1,1) and (PP.power, PP.coloc) less than .25) for colocalization analysis is not widely used, and a more typical threshold of PP.coloc > 0.5 or 0.8 should be used.

5. The colocalization analysis between sQTLs and eQTLs from GTEx should be provided even though the overlap is marginal.

Minor comments:

1. Fig 1B. “Significant genes” for x-axis should be “Genes with a significant ψQTL”

2. The following text should be moved to the discussion section:

“While Leafcutter [8] identifies more splicing events and finds more sQTLs, it presents an interpretability challenge. It is often difficult to identify which exon a Leafcutter cluster corresponds to, and effect directions are sometimes unclear. While ψQTLs are less powerful in a statistical sense, the method clearly links splicing events to exons, genesgenes, and effect directions, which was advantageous to the purpose of this study.

As an additional follow up, we performed ψQTL mapping in the Geuvadis [30]dataset, and checked for concordance with GTEx EBV-transformed lymphocytes. (See methods for details) Of the 1,119 sGenes with coverage in Geuvadis and GTEx, 423 had a significant sExon in both. The coverage in Geuvadis was smaller overall, only capturing splicing for 853 out of 1431 exons with a significant ψQTL in GTEx lymphocytes. Among this set, however, the concordance was reasonably high, with a π1 score of 0.63. This replication strengthens evidence that ψQTLs are robust in many contexts.”

3. “p < 2e-16” is a default minimum p value. Please use the true p value.

4. When the Fisher’s Exact test is applied, it is not clear what is the background used? For example, what is the background in the following test:

“We found that indeed, among sExons, 41.20% were symmetric compared 157 to 38.77% of all non-terminal exons annotated in gencode v26 (Fig 1C, Fisher’s Exact Test p = 6.64x10-4). ”

Reviewer #2: The authors have addressed all concerns. One additional suggestion would be to include a supplemental table with the locations and variant ID associated with all of the ∼4k psi-QTLs/sExon's analyzed in the study, so that other scientists can easily work with the same data.

7. PLOS authors have the option to publish the peer review history of their article (what does this mean?). If published, this will include your full peer review and any attached files.

Reviewer #1: No

Reviewer #2: No

---

## [Author Response · Author response to Decision Letter 1]

19 Jul 2023

Reviewer #1: In the Einson et al. manuscript, the authors identified splicing quantitative trait loci (sQTLs) using exon percent spliced in (PSI) scores as a quantitative phenotype. They also utilized AlphaFold2 to predict changes in protein structure associated with sQTLs that overlapped with GWAS traits, highlighting a potential new use-case for this technology in interpreting genetic effects on traits and disorders. This study presents novel insights into the effects of sQTLs on splicing events and GWAS signals.

However, there are some major comments to be addressed.

1. The study lacks a direct comparison with those protein coding genes with sQTLs from previously published studies, such as the one by Garrido-Martín et al. (2021)

Garrido-Martín, D., Borsari, B., Calvo, M. et al. Identification and analysis of splicing quantitative trait loci across multiple tissues in the human genome. Nat Commun 12, 727 (2021). https://doi.org/10.1038/s41467-020-20578-2

RE: We had reported the comparison of our sQTL calls to the GTEx v8 sQTL calls based on LeafCutter, which is the most commonly used method. However, we agree that a comparison to the sQTLs from the referenced paper is valuable. Thus, we obtained the sQTL calls from Garrido-Martin et al., and now report the comparison to these in line 133, and Supplementary Figure 1E. 

2. It is not clear which exon is considered the "top exon" in the statement "removing genes where the 3' or 5' terminal exon was the top exon." If the top exon refers to the first or last exon, then the majority of genes would be removed since most genes have a 3' or 5' exon.

RE: Thank you for raising this point of confusion. Here, “top exon” refers to the exon whose splicing is most significantly associated with a genetic variant. This has been updated in the text (line 137)

3. While the annotations (e.g., exonic, intronic, or intergenic) of sVariants are provided, there is no information about whether an sVariant is deleterious or benign.

RE: The simple annotation is provided for descriptive purposes, and we do not assign deleteriousness scores to sVariants, because this analysis would be unlikely to yield informative and interpretable results: Firstly, the majority of them are noncoding, and accurate deleteriousness scores are available only for coding variants, and biologically informative only when the sVariant is the true causal variant rather than LD proxy. Secondly, because sVariants are relatively common in the population and are associated with common changes in alternative splicing, they are unlikely to be severely damaging. 

4. The threshold (i.e. Euclidean distance between (1,1) and (PP.power, PP.coloc) less than .25) for colocalization analysis is not widely used, and a more typical threshold of PP.coloc > 0.5 or 0.8 should be used.

RE: As explained in line 560, we chose this looser definition of colocalization to allow for more data in downstream analysis, and where the false positive rate is less critical. Originally, we had implemented a strict cutoff of PP.coloc > 0.8, but this is highly conservative and produces many false negatives, and implementing that substantially limited our power to detect associations with colocalization status and splicing change. There is no systematic reason that a looser threshold with potentially same false positive colocalizations would bias the patterns or outcomes that we report. We have added a note of this in Methods advising caution in interpretation of loci where the evidence for colocalization is not very strong. 

5. The colocalization analysis between sQTLs and eQTLs from GTEx should be provided even though the overlap is marginal.

RE: Thank you for this suggestion. We have included the credible sets overlaps between sQTLs and eQTLs in the Zenodo repository. It is now live with the new version. 

Minor comments:

1. Fig 1B. “Significant genes” for x-axis should be “Genes with a significant ψQTL”

RE: Thank you for the comment. The axis label of this figure has been updated. 

2. The following text should be moved to the discussion section:

“While Leafcutter [8] identifies more splicing events and finds more sQTLs, it presents an interpretability challenge. It is often difficult to identify which exon a Leafcutter cluster corresponds to, and effect directions are sometimes unclear. While ψQTLs are less powerful in a statistical sense, the method clearly links splicing events to exons, genes, and effect directions, which was advantageous to the purpose of this study.

As an additional follow up, we performed ψQTL mapping in the Geuvadis [30]dataset, and checked for concordance with GTEx EBV-transformed lymphocytes. (See methods for details) Of the 1,119 sGenes with coverage in Geuvadis and GTEx, 423 had a significant sExon in both. The coverage in Geuvadis was smaller overall, only capturing splicing for 853 out of 1431 exons with a significant ψQTL in GTEx lymphocytes. Among this set, however, the concordance was reasonably high, with a π1 score of 0.63. This replication strengthens evidence that ψQTLs are robust in many contexts.”

RE: We think that the overall clarity of the manuscript would suffer if this text was moved. The description of the LeafCutter comparison presents a rationale for the work, and moving this to the end could leave a reader wondering why we did not just use the GTEx sQTLs as is. This topic is elaborated further in Discussion. The replication analysis reports results of an analysis, which in our opinion belongs to Results. 

3. “p < 2e-16” is a default minimum p value. Please use the true p value.

RE: Thank you for this comment. We have updated statistics to the true p value at line 148, twice at line 269, at line 285 and at line 286. Figure 3 has also been updated accordingly. 

4. When the Fisher’s Exact test is applied, it is not clear what is the background used? For example, what is the background in the following test:

“We found that indeed, among sExons, 41.20% were symmetric compared to 38.77% of all non-terminal exons annotated in gencode v26 (Fig 1C, Fisher’s Exact Test p = 6.64x10-4). ”

RE: Thank you for pointing out this confusing language. We have clarified the background in the relevant sections. In the analysis pointed out here, we are comparing sExons to non-sExons (exons without a significant ψQTL), and stratifying them by being symmetric or not. 

Reviewer #2: The authors have addressed all concerns. One additional suggestion would be to include a supplemental table with the locations and variant ID associated with all of the ∼4k psi-QTLs/sExon's analyzed in the study, so that other scientists can easily work with the same data.

RE: This information is already included in the file top_sQTLs_MAF05.tsv, which is located in the 02_qtl_results subdirectory of the Zenodo page. If researchers need rsIDs, they can convert them from the top_vid column.

---

## [Decision Letter · Decision Letter 2]

10 Sep 2023

The impact of genetically controlled splicing on exon inclusion and protein structure

PONE-D-23-00371R2

Dear Dr. Einson,

We’re pleased to inform you that your manuscript has been judged scientifically suitable for publication and will be formally accepted for publication once it meets all outstanding technical requirements.

Kind regards,

Chunyu Liu

Academic Editor

PLOS ONE

Additional Editor Comments (optional):

Reviewers' comments:

Reviewer's Responses to Questions

**Comments to the Author**

1. If the authors have adequately addressed your comments raised in a previous round of review and you feel that this manuscript is now acceptable for publication, you may indicate that here to bypass the “Comments to the Author” section, enter your conflict of interest statement in the “Confidential to Editor” section, and submit your "Accept" recommendation.

Reviewer #1: All comments have been addressed

2. Is the manuscript technically sound, and do the data support the conclusions?

Reviewer #1: Yes

3. Has the statistical analysis been performed appropriately and rigorously? 

Reviewer #1: Yes

4. Have the authors made all data underlying the findings in their manuscript fully available?

Reviewer #1: Yes

5. Is the manuscript presented in an intelligible fashion and written in standard English?

Reviewer #1: Yes

6. Review Comments to the Author

Reviewer #1: (No Response)

7. PLOS authors have the option to publish the peer review history of their article (what does this mean?). If published, this will include your full peer review and any attached files.

Reviewer #1: **Yes: **Xusheng Wang

---

## [Editor Report · Acceptance letter]

15 Sep 2023

PONE-D-23-00371R2 

The impact of genetically controlled splicing on exon inclusion and protein structure 

Dear Dr. Einson:

I'm pleased to inform you that your manuscript has been deemed suitable for publication in PLOS ONE. Congratulations! Your manuscript is now with our production department. 

Kind regards, 

on behalf of

Dr. Chunyu Liu 

Academic Editor

PLOS ONE